# How Out-of-Distribution important is

## Abstract

Class Incremental Learning (CIL) has gained significant attention in recent years due to its potential to adaptively learn from a non-stationary data distribution. The challenge of CIL primarily revolves around the model's ability to learn new classes without forgetting previously acquired knowledge. Recent research trends has achieved significant milestones, yet the continuity of learning can be further strengthened by integrating the concepts of "self-training", "out-of-distribution", and "data drift". In this paper, we propose a novel approach that integrates "Continual Learning", "Self-Training", "Out-of-Distribution recognition", and "Data Drift" concepts to advance the capabilities of class incremental learning systems. Drawing inspiration from works such as "A Theoretical Study on Solving Continual Learning (Kim et al., 2022)", and "CSI: Novelty Detection via Contrastive Learning on Distributionally Shifted Instances (Tack et al., 2020)". We propose a model that satisfies the four concepts mentioned above. Our experimental results demonstrate the efficacy of this method in mitigating catastrophic forgetting and ensuring consistent performance across a diverse range of classes.

## 1 Introduction

The realm of machine learning and deep learning has always thrived on the idea of adaptation and evolution. As more real-world applications demand on-the-fly learning capabilities from models, the paradigm of Continual Learning (CL) stands out. One of the popular task is Class Incremental Learning (CIL). CIL embodies the idea of a model's ability to incrementally learn new classes as data becomes available, rather than having to be retrained from scratch or with a fully available dataset. This paradigm aligns well with real-world scenarios, where data often comes in streams or batches, and there's a constant need for the model to update its knowledge without compromising its previously learned tasks.

However, the journey of CIL has not been without its challenges. A significant issue plaguing CIL is catastrophic forgetting, where neural networks tend to forget previous knowledge while accommodating new information. Moreover, in real-world scenarios, data often does not adhere to fixed distributions. Over time, the nature and distribution of the data might shift, a phenomenon commonly referred to as "data drift". Similarly, "out-of-distribution" data points, which are not part of the training data distribution, can throw a wrench into the smooth functioning of models. The integration of "self-training", where models use their predictions to further train themselves, has emerged as a promising avenue to address some of these concerns.

Several studies, such as CSI (Tack et al., 2020) and "RECL: Responsive Resource-Efficient Continuous Learning for Video Analytics (Khani et al., 2023)", have made significant contributions to this field. They have paved the way for the current research landscape, opening up opportunities to integrate disparate yet crucial concepts like "self-training", "out-of-distribution", and "data drift" into the CL framework.

In light of these developments, this paper aims to propose a consolidated approach that not only addresses the inherent challenges of CIL but also synergizes the aforementioned concepts to achieve a more robust and adaptive CIL system. Through experiments and evaluations, we seek to underline the advantages of this integrated approach, hoping to set a new benchmark in the domain of class incremental learning.

Moreover, the methodology we introduce holds profound implications for anomaly detection. Especially in applications like wafer defect detection, where not only the intricacy of identifying defects

matters, but also accounting for data drift is paramount. As defect patterns might evolve or change over time, models must be adept at identifying these novel patterns without any lapse in their primary detection capability.

## 2 RELATED WORKS

The path toward achieving robust Class Incremental Learning (CIL) systems has been paved by a myriad of research works, addressing various facets of continual learning, self-training, out-of-distribution recognition, and data drift. In this section, we delve into some of the seminal works and recent advancements that have laid the foundation for our proposed methodology.

### 2.1 CONTINUAL LEARNING

One of the core issues that CL tries to address is catastrophic forgetting. The previous work (Kim et al., 2022) delves deep into the underlying causes of this phenomenon and suggests several strategies for mitigation. Intriguingly, this study has also demonstrated a noteworthy relationship between out-of-distribution (OOD) prediction performance and CIL performance, indicating that they are proportionally related. This observation underscores the intertwined challenges of handling OOD samples while maintaining continuity in learning. The paper emphasizes the importance of synaptic consolidation, where certain weights in the neural network are preserved to maintain knowledge of previously learned tasks. Such strategies, combined with the insights on OOD and CIL relationship, form the backbone of many state-of-the-art CL systems and have significantly influenced our proposed architecture.

### 2.2 SELF-TRAINING

Self-training is a paradigm wherein a model harnesses its own predictions, typically those with high confidence, to further refine and improve its performance. The premise behind this is that high-confidence predictions can serve as pseudo-labels, allowing the model to expand its training data iteratively. While not directly related to CIL, the principles of self-training can be integrated to enhance the model's robustness, especially when new classes of data are introduced without ground-truth labels.

### 2.3 OUT-OF-DISTRIBUTION RECOGNITION

Handling out-of-distribution (OOD) samples is crucial for any model meant to be deployed in real-world settings. Such data points, which lie outside the training distribution, can lead to unpredictable and often incorrect predictions. The work, CSI (Tack et al., 2020), provides a novel approach to tackle this issue. By employing contrastive learning, the authors are able to differentiate between in-distribution and OOD samples effectively. Another significant contribution in this area comes from other paper (Fang et al., 2022), which has shown that OOD learning is possible under certain conditions. This supports that our study is OOD learning-capable and has marked effects, especially in the Object Detection. Such insights can be immensely beneficial for CIL, as new classes might often introduce OOD samples that the model hasn't previously encountered.

### 2.4 DATA DRIFT

Data drift refers to the change in data distribution over time. This is particularly relevant in dynamic environments where the statistical properties of the input data can change. RECL (Khani et al., 2023) underscores the challenges posed by such drifts, especially in video analytics, where the temporal nature of the data introduces unique challenges. The RECL approach, which dynamically adjusts resources based on the changing nature of data, hints at possible strategies that can be employed in CIL systems to account for data drift.

### 2.5 ANOMALY DETECTION

Beyond the conventional challenges of CIL, anomaly detection stands out as a pertinent application. Especially in contexts like wafer defect detection, where the data can not only drift but also introduce

novel defect patterns. The paper "Deformable Convolutional Networks for Efficient Mixed-Type Wafer Defect Pattern Recognition (Wang et al., 2020)" provides a compelling insight into this area. By leveraging deformable convolutional networks, the authors introduce a flexible approach to recognize a variety of defect patterns, particularly in mixed-type wafer scenarios. Such methodologies emphasize the need for adaptive neural architectures in the face of dynamic and diverse anomalies. Integrating these insights, our proposed model strives to be adept at recognizing these anomalies amidst the continual introduction of new classes, thereby ensuring the integrity and reliability of the detection mechanism.

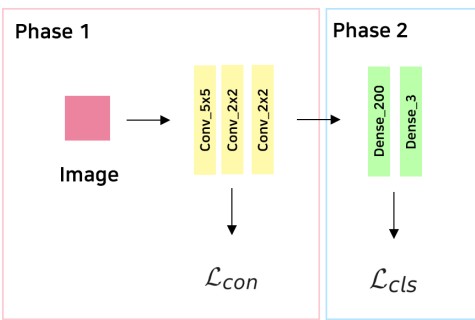

Figure 1: Image of the model architecture and explanation of 2-phase learning. In Phase 1, feature extractor is learned through contrastive learning using contrastive loss, and in Phase 2, classifier is learned through classification learning

## 3 MAIN CONCEPT

In the subsequent sections, we delve deep into the architecture of the Data Drift Detect Model.

### 3.1 DATA DRIFT DETECT MODEL

Drawing inspiration from prior works (Kim et al., 2022), we learned that the ability to detect Out-of-Distribution (OOD) can bolster both CIL and TIL. This ability to specify unseen data allows our model to categorize such data as a new class and subsequently self-train. Though there are concerns that labeling unseen data as a new class could lead to inappropriate behavior, in constrained scenarios (e.g., wafer defect detection), such an ability becomes instrumental for continuous learning and self-training.

Our foundational architecture is influenced by CSI (Tack et al., 2020). However, we questioned the efficacy of assigning a class of out-of-distribution to four times the number of typical classes. Instead, we designed our model to have a singular, significantly reduced OOD class. Our approach uses a two-phase learning methodology: first, the feature extractor is trained via contrastive learning, and then the classifier is refined through classification learning, as illustrated in Figure 1.

Now, let's delve into the learning flow, which can be segmented into five distinct steps, with steps 3 to 5 being iterative. This explanation is particularly tailored for the Anomaly Detection model, so it might vary slightly from conventional CIL settings.

For better clarity, it's recommended to refer to Figure 3 while navigating through the subsequent steps:

- **Step 1:** Starting with normal data and two sets of abnormal data, we initiate training using data augmented from the abnormal set. In this phase, the first abnormal data is labeled as 'defect' and the remainder is tagged as OOD. The resulting trained model is termed as 'model1'.

- **Step 2:** Similarly, with normal data, two sets of abnormal data, and the augmented set, we now classify the second abnormal data as 'defect', relegating the remainder to the OOD category. The model derived from this training is designated 'model2'.

---

**Algorithm 1** Model inference flow (Defect Detection)

---

1: **for** stream data **do**
2:     $R \leftarrow -1$
3:     **for** model set **do**
4:         $r \leftarrow m(x)$
5:         **if** $r == 0$ **then**
6:             $R \leftarrow 0$
7:             $break$
8:         **else if** $r == 1$ **then**
9:             $R \leftarrow 1$
10:             $break$
11:         **end if**
12:     **end for**
13:     **if** $R == -1$ **then**
14:         $ds \leftarrow ds + x$
15:         **if** $len(ds) \geq \theta$ **then**
16:             $train(ds)$
17:         **end if**
18:     **end if**
19: **end for**

---

- **Step 3:** The two pre-trained models (model1 and model2) are now deployed for stream data classification. As data evolves over time, introducing new class anomalies, our system detects such anomalies as data drifts, classifying them as 'unknown defect'.

- **Step 4:** Using normal data, two abnormal datasets, and the freshly identified 'unknown defect' data, a new augmented set is prepared. In this phase, 'unknown defect' data is classified as 'defect', and any remaining anomalies are classified as OOD. The ensuing model from this phase is labeled 'model3'.

- **Step 5:** Here, the stored 'unknown defect' data is trained alongside new defects, continually refining the model.

The contents of steps 3 to 5 can be further identified in Algorithm 1, where the loop runs for stream data and stores -1 in the final result $R$. Then, we run inference while circulating the model set (or parallel computation is also possible), and if the result $r$ is 0 or 1, we store 0 (normal) or 1 (abnormal) in the final result and end it. However, if the end result is -1 even though all models have finished inference, this means that all models have inferred as OOD and add that data $x$ to the new Dataset, $ds$. At that time, if the length of the $ds$ is greater than the predetermined theta value, the new model is trained using the collected data.

We found through implementation and experiments that these model designs can be used not only for image classification but also for data drift detection, continuous learning, and self-training to maintain accuracy in the field of object detection. Figure 2 shows that there are Center Cluster and Edge Cluster, and if you look inside, there are Object Detection models and Drift Detection models. This separated structure was inspired by Fang et al. (2022). In addition, stream video data is received through an ip camera, and at this time, the model proceeds with Object Detection, and if something is detected as a result, Drift Detection is performed, and if a data drift occurs, data is sent to the Center Cluster to fine-tune the model. By using this system, we can see that we have created a system that adapts to environmental changes.

## 4 EXPERIMENT

### 4.1 WAFER DEFECT DETECTION

We utilized a subset of the MixedWM38 dataset (Wang et al., 2020), with some modifications. As illustrated in Figure 4, the dataset encompasses seven classes: Normal, Center Defect, Donut Defect, Edge-loc Defect, Edge-ring Defect, Loc Defect, and Scratch Defect. The initial learning process only encompasses the Normal, Center, and Donut Defect classes. Following this, an actual

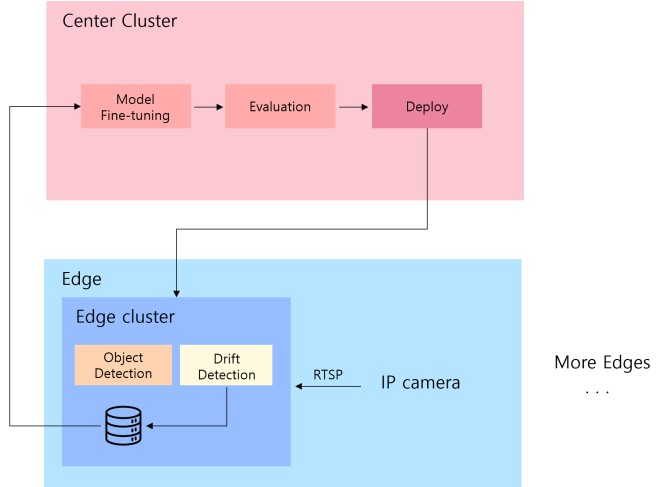

Figure 2: The MLOps Pipeline image of Object Detection Model with self-training to adapt to Data Drift that may occur over time. Center Cluster means Kubernetes cluster on cloud servers and Edge Cluster means Kubernetes cluster on embedded boards. The model is used in combination with the Object Detection model, and when data Drift is detected, image data and results detected by the Object Detection model are stored as label data, and when data is accumulated above a certain threshold, data is sent to the Center Cluster to start model Fine tuning.

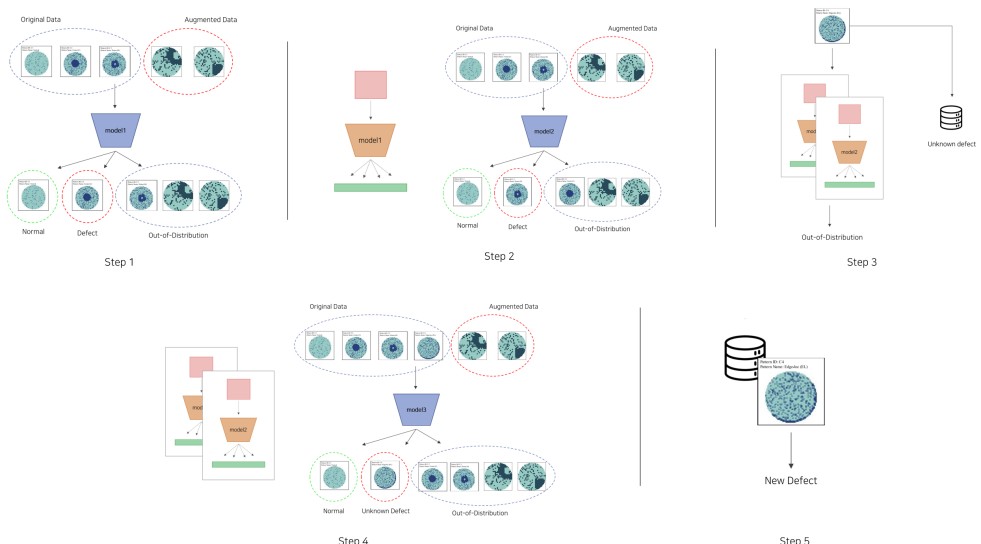

Figure 3: Depiction of the model's learning flow. The process can be divided into five core steps, with steps 3 to 5 being recurrent. Smaller, specialized models handle individual classes and amalgamate to shape a holistic classifier. Refer to the main text for an in-depth explanation.

distribution simulation is executed. Initially, 3,000 data points from the initially learned classes are introduced, followed by the introduction of new classes at intervals of 2,000.

Initially, the encoder is learned using Contrastive learning. This encoder only exists and learns anew whenever a new class is added, which is used jointly by all models. At this time, the learning rate proceeds with 0.001, batch size 16, dropout rate 0.25 and epoch 5.

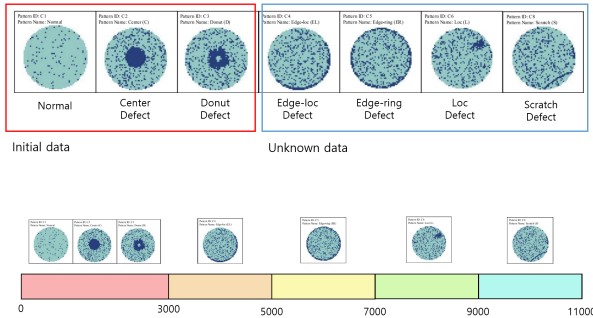

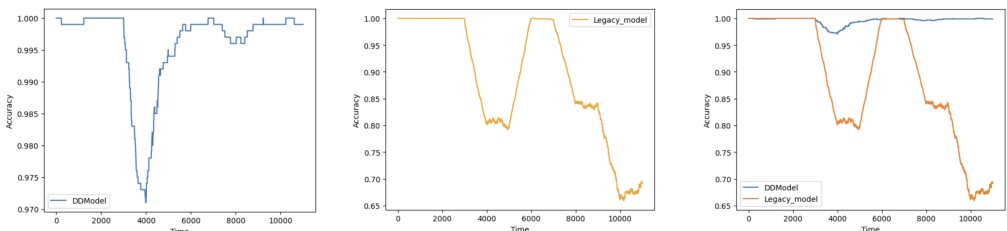

Figure 4: This is an image of the MixedWM38 (Wang et al., 2020) data used in Wafer Defect Detection, showing seven classes on the top, three on the left, and unknown data on the right, stream data to be used in simulation on the bottom, showing initial data, and then creating new classes at intervals of 2,000.

Subsequently, after constructing an additional classifier as many classes as possible, learn using classification learning. At this time, all of the weight values of the encoder are fixed, and only the classifier is learned. At this time, the learning rate proceeds with 0.001, batch size 16, dropout rate 0.5 and epoch 5.

Thereafter, 11,000 stream data are built and simulations are started to infer one by one, and if the data results in OOD, the data is loaded into memory, and if the number of loaded data exceeds 300, the new model is learned. At this time, the learning flow proceeds in the same two-phase form as in the beginning, and at this time, the weight of the encoder has been updated, so the classifier of all other existing models also learning again. Hyperparameters are the same as before.

During the simulation, the accuracy is recorded continuously, and the accuracy is recorded as the average accuracy of the previous 1000 inferences. And the initial accuracy starts with 1.0.

Since we need a target model to compare and analyze with our model, we separately build a legacy model without the self-training method, and the layer architecture of that model has the same structure and applies the 2-phase learning of Contrastive and Classification equally.

Figure 5: Accuracy graph of the Wafer Defect Detection Simulation. Data Drift model and legacy model are used. See the body for more detail explanation.

The results are shown in Figure 5, and it can be seen that there is a large difference between the self-training model (DDModel) and the legacy model.

In the initial 0-3000 section, both models show good accuracy, but in the 3000 section where new classes appear, both models show a decrease in accuracy. However, it should be noted that the two models with the same structure and conducted the same hyperparameter and two-phase learning have different slopes with reduced accuracy. This shows that DDModel is inherently capable of emerging a new class inside because it is trained to detect OOD as well as normal and abnormal with augmented data at the time of learning. This also shows to some extent that the accuracy of the DDModel decreased slightly in the 0 to 2000 sections.

It does not appear in the graph, but during the simulation, the DDModel begins self-training on the 3616th data Given that there are about 308 new data up to the 3616th, 300 of them were identified by OOD, so almost all new data were identified. After self-training, it can be seen that the DDModel recovers its accuracy. However, the legacy model maintains relatively low accuracy up to about 5000th data, and in the 5000-7000 section, almost all data are matched, which means that the legacy model we learned is not overfit for the initial data, but has the so-called "generalization" ability. However, from the 9000th data, the accuracy decreases again.

DDModel shows good accuracy after completing first self-training, but shows a slight decrease in accuracy as it enters a new class from the 7000th data, but it shows a self-training process again to increase accuracy The DDModel was shown to complete the simulation while maintaining high accuracy through a total of three self-training sessions.

## 4.2 GENERAL CLASS INCREMENTAL LEARNING

Traditional Class Incremental Learning (CIL) approaches often follow a well-structured path wherein each class or group of classes is learned sequentially. However, our proposed model takes a distinct approach, particularly when it's compared to other continual learning methodologies, which presents some inherent challenges in benchmarking performance.

For instance, using the MNIST dataset as a representative example, the general flow for many CIL models would be to sequentially learn classes: first classes 0 and 1, followed by classes 2 and 3, and so on. In this setup, during the learning of new classes, the model has prior knowledge about the labeling of these classes. On the contrary, our model doesn't have the advantage of labeled data for any classes beyond the initial set. This disparity renders a direct comparison between our model and traditional models potentially unfair. Our model is handicapped in a sense, due to the lack of access to labeled data.

However, the importance of quantitative performance benchmarking is undeniable. To this end, while maintaining the overarching framework of our methodology, we introduced specific modifications to bring it closer to conventional CIL settings for comparison purposes. We removed the Replay Buffer, allowing the model to learn about other classes in advance. The restructured learning flow and model architecture can be visualized in Figure 6, which elucidates that instead of the prior five-step process, the learning has been condensed to just three major steps.

- **Step 1:** We commence with the digits 0 and 1. The dataset comprising these two numbers is augmented. After this augmentation process, the model is trained to distinguish between 0, 1, and OOD. The resulting model post-training at this phase is referred to as 'Model 1'.
- **Step 2:** Next, digits 2 and 3 are taken. Augmented data is created similar to the prior step. The training proceeds, distinguishing between 2, 3, and OOD. The model post this training is termed 'Model 2'.
This process is repetitively executed until we train the model with digits 8 and 9, resulting in a sequence of models from 'Model 0' to 'Model 5'.
- **Step 3:** During the inference phase, the accumulated models ('Model 1' to 'Model 5') are utilized. The inference methodology can be comprehended in detail by referring to algorithm 2.

Algorithm 2 conducts inference on the stream data that comes in to explain the inference process in Step 3 described above. For each data, -1 is stored in $R$, which represents the result, and $S$, which represents the score, and inference is carried out while circulating the model (or may be executed in parallel). The model receives data $x$ and then outputs $s1$, $s2$, and $s3$, which are expressed as probabilities of the first class, the second class, and the OOD class, respectively. We then compare $s1$ and $s2$ to $S$, respectively, and if they are larger, we use the index of the model to get the correct answer, save it, and update the score to that score.

The results of our model modified in this way can be seen in Table 1, we observed intriguing results. When benchmarked against traditional CIL models, our approach did exhibit some performance degradation, primarily attributable to the absence of the Replay Buffer and the unfamiliarity with unseen classes. However, it's noteworthy that the robustness against OOD and adaptability of the model remained steadfast, indicating its versatility across varied operational scenarios.

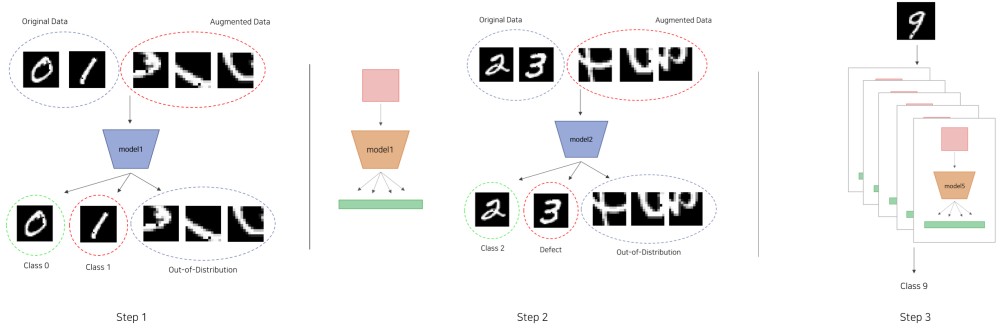

Figure 6: Depiction of the model's learning flow. The process can be divided into three core steps. Smaller, specialized models handle two classes and amalgamate to shape a holistic classifier. Refer to the main text for an in-depth explanation.

---

**Algorithm 2** Model inference flow (MNIST CIL)

---

1: **for** stream data **do**
2:    $R \leftarrow -1$
3:    $S \leftarrow -1$
4:    **for** model set with idx **do**
5:       $s1, s2, s3 \leftarrow m(x)$
6:       **if** $S < s1$ **then**
7:          $R \leftarrow idx * 2$
8:          $S \leftarrow s1$
9:       **else if** $S < s2$ **then**
10:         $R \leftarrow idx * 2 + 1$
11:         $S \leftarrow s1$
12:       **end if**
13:    **end for**
14: **end for**

---

Table 1: Table for MNIST score

| Method | M-5T |
|---|---|
| MUC | 74.9 |
| PASS | 76.6 |
| HAT | 81.9 |
| HyperNet | 56.6 |
| Sup | 70.1 |
| PR-Ent | 74.1 |
| Sup+CSI | 80.7 |
| DDModel | **85.2** |

In addition, we conducted an experiment on whether it would be unconditionally good to design a model that classifies Class and a model that detects OOD separately (Fang et al., 2022).

All other conditions were kept the same and the number of classifiers was increased to two, one was used as a classifier for classifying Class and the other was used as a classifier for classifying OOD. Accordingly, both output sizes take the form of binary classifiers.

The result was 85.26 to 47.46 that the model was separated and built was much worse. Accordingly, we can see that there is no correct answer to continuous learning and that several methodologies must be used according to Task.

## 5 CONCLUSION

Our research embarked on the ambitious endeavor of amalgamating the concepts of Continual Learning, Self-Training, Out-of-Distribution recognition, and Data Drift to devise a comprehensive model for Class Incremental Learning. Our architectural choices were underscored by a two-phase learning methodology, emphasizing both feature extraction and classification refinement. The experimental validations, particularly the Wafer Defect Detection and General Class Incremental Learning, reinforced the efficacy of our approach in dynamically identifying new classes and refining learning without significant overhead.

Moreover, the model's resilience against Out-of-Distribution instances and its dexterity in data drift detection fortify its potential in real-world, non-stationary data environments. The critical distinction between our model and traditional CIL models lies in its ability to handle unknown classes, making it a versatile tool for scenarios where data continuously evolves.

## 6 LIMITATIONS

While our proposed model presents advances in the realm of Class Incremental Learning, it is not without its limitations:

- **Scalability**: As the number of classes or the volume of data increases, the iterative refinement processes might become computationally intensive.
- **Comparative Benchmarking**: Direct comparison with traditional CIL models, due to our unique approach, could be perceived as skewed or incomprehensive in certain contexts.

## 7 FUTURE WORK

In light of our findings and the recognized limitations, several avenues are ripe for exploration:

- Exploring computational optimizations, potentially leveraging distributed computing or hardware accelerations, to enhance scalability.
- Adapting the model to function optimally with fewer data, venturing into semi-supervised or unsupervised territories.
- Designing comprehensive benchmarking strategies, to assess the model's performance across varied continual learning paradigms.

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
