# OpenReview forum: "How Out-of-Distribution important is"
_ICLR.cc/2024/Conference — Submitted to ICLR 2024_

### Official Review · Reviewer_6xad · 2023-10-14

**Soundness:** 1 poor
**Presentation:** 1 poor
**Contribution:** 1 poor
**Rating:** 1
**Confidence:** 3

**Summary:**

The presentation is poor such that I can not understand what are the true purposes of this paper and I can not catch the key contributions of this paper.

**Strengths:**

Can not answer.

**Weaknesses:**

1. The presentation is poor to understand what points this paper want to show?

2. The title of this paper is named "How out-of-distribution" important is" but I can not see any support to answer this question.

3. I can not catch the contributions of this paper due to the poor undertanding.

**Questions:**

Rewriting this paper is necessary to emphasize the theme and contributions.

---

### Official Review · Reviewer_fB3F · 2023-10-22

**Soundness:** 3 good
**Presentation:** 3 good
**Contribution:** 2 fair
**Rating:** 3
**Confidence:** 4

**Summary:**

This paper aims to propose a consolidated approach that not only addresses the inherent challenges of CIL but also synergizes the aforementioned concepts to achieve a more robust and adaptive CIL system.

**Strengths:**

This paper presents a clear framework and the framework takes Continual Learning, Self-Training, Out-of-Distribution recognition, and Data Drift into consideration.

**Weaknesses:**

The style of this paper is not suitable for ICLR, the LEARNING part is not sufficient, it is suitable for pitching some data mining conferences.

**Questions:**

1. With respect to the design of the OOD component, what is specifically better about the framework in this paper compared to CSI?

2. How is the augmentation mentioned in the text portrayed, some augmentation can't actually be utilized as OOD.

3. This approach requires 3 models. Does this bring more training overhead while improving results?

**Details Of Ethics Concerns:**

None.

---

### Official Review · Reviewer_DuXE · 2023-11-01

**Soundness:** 2 fair
**Presentation:** 1 poor
**Contribution:** 2 fair
**Rating:** 3
**Confidence:** 4

**Summary:**

This paper proposes a class-incremental learning method that integrates Continual Learning, Self-Training, Out-of-Distribution recognition, and Data Drift concepts.

**Strengths:**

Exploring the effectiveness of Out-of-Distribution recognition for improving class-incremental learning is an valuable direction.

**Weaknesses:**

1. The proposed method is not clear, making it difficult to follow. For example, the formulation of the learning objective is missing. What kind of data augmentation is used in Figure 3, and where is the abnormal set from?
2. The experiment is poor, where only Table 1 on MINST is presented. Moreover, the details of implementation are missing.
3. The references of compared methods in Table 1 are missing, and many other related works of continual learning are missing.

**Questions:**

Which network is used to perform experiments?

---

### Official Review · Reviewer_SGSA · 2023-11-01

**Soundness:** 3 good
**Presentation:** 1 poor
**Contribution:** 2 fair
**Rating:** 3
**Confidence:** 3

**Summary:**

The authors address an important problem, integrating continual learning with self-training, OOD detection, and data drift. They propose a learning scenario that outlines a series of learning steps for an AI system under an existence of OOD samples, distribution shift, and continual training. The proposed method is test with wafer defect detection and MNIST classification

**Strengths:**

The paper tries to solve an important problem

**Weaknesses:**

1. The writing is not clear in several sections. For instance, the authors mention that their work is inspired from CSI, but the authors did not provide sufficient details about the CSI method. From the current manuscript, my understanding of CSI limited to it being an out-of-distribution (OOD) detection paper.

2. It's not clear what the proposed method is and how it's implemented.

3. The experiment setups are not clear (e.g., task split or how OOD data are provided.)

4. The experiment is too simple to assess the significance. The method is only evaluated on the wafer defect detection and MNIST classification.

Misc. comments; Please use `` instead of " to open a quotation.

**Questions:**

See the Weaknesses

---

### Meta-Review · Area_Chair_eVDK · 2023-12-01

**Metareview:**

This paper describes an approach to integrating continual learning with out-of-distribution data detection and self-training (which seems like a type of pseudolabeling in the author interpretation). While the paper addresses an interesting and important area, the paper does not lay out a clear motivation for the combination self-training and OOD detection, nor is there a coherent description of the proposed approach. Moreover, the related work section is quite sparse, citing only a single continual learning paper and a single OOD detection paper. Finally, the experimental results are given on a severely limited range of datasets and scenarios. The paper seems to be an early draft of what might become an interesting approach, but it is far to immature for publication at this time.

**Justification For Why Not Higher Score:**

The reviewer consensus is unanimous that this paper lacks clarity and clear statement (and support for) a central thesis. The authors did not respond to the reviewer comments.

**Justification For Why Not Lower Score:**

N/A

---

### Decision · Program_Chairs · 2024-01-16

Reject